# The M35 Metalloprotease Effector FocM35_1 Is Required for Full Virulence of *Fusarium oxysporum* f. sp. *cubense* Tropical Race 4

**DOI:** 10.3390/pathogens10060670

**Published:** 2021-05-29

**Authors:** Xiaoxia Zhang, Huoqing Huang, Bangting Wu, Jianghui Xie, Altus Viljoen, Wei Wang, Diane Mostert, Yanling Xie, Gang Fu, Dandan Xiang, Shuxia Lyu, Siwen Liu, Chunyu Li

**Affiliations:** 1College of Bioscience and Biotechnology, Shenyang Agricultural University, Shenyang 110866, China; zhangxiaoxia418@163.com; 2Key Laboratory of South Subtropical Fruit Biology and Genetic Resource Utilization, Ministry of Agriculture, Key Laboratory of Tropical and Subtropical Fruit Tree Research of Guangdong Province, Institution of Fruit Tree Research, Guangdong Academy of Agricultural Sciences, Guangzhou 510640, China; hqhuang07@163.com (H.H.); bangtingwu@163.com (B.W.); xieyanling19@webmail.hzau.edu.cn (Y.X.); xiangdandan@gdaas.cn (D.X.); lichunyu@gdaas.cn (C.L.); 3Key Laboratory of Biology and Genetic Resources of Tropical Crops, Ministry of Agriculture, Institute of Tropical Bioscience and Biotechnology, Chinese Academy of Tropical Agricultural Sciences, Haikou 571101, China; xiejianghui@itbb.org.cn (J.X.); wangwei@itbb.org.cn (W.W.); 4Department of Plant Pathology, University of Stellenbosch, Matieland 7602, South Africa; altus@sun.ac.za (A.V.); diane@sun.ac.za (D.M.); 5Institute of Plant Protection, Guangxi Academy of Agricultural Sciences, Guangxi Key Laboratory of Biology for Crop Diseases and Insect Pests, Nanning 530007, China; fug110@gxaas.net

**Keywords:** Fusarium wilt, banana, virulence, effector, metalloprotease

## Abstract

*Fusarium oxysporum* f. sp. *cubense* tropical race 4 (*Foc* TR4) causes Fusarium wilt of banana, the most devastating disease on a banana plant. The genome of *Foc* TR4 encodes many candidate effector proteins. However, little is known about the functions of these effector proteins on their contributions to disease development and *Foc* TR4 virulence. Here, we discovered a secreted metalloprotease, FocM35_1, which is an essential virulence effector of *Foc* TR4. FocM35_1 was highly upregulated during the early stages of *Foc* TR4 infection progress in bananas. The FocM35_1 knockout mutant compromised the virulence of *Foc* TR4. FocM35_1 could interact with the banana chitinase MaChiA, and it decreased banana chitinase activity. FocM35_1 induced cell death in *Nicotiana benthamiana* while suppressing the INF1-induced hypersensitive response (HR), and its predicted enzymatic site was required for lesion formation and the suppression to INF1-induced HR on *N. benthamiana* leaves. Importantly, treatment of banana leaves with recombinant FocM35_1 accelerates *Foc* TR4 infection. Collectively, our study provides evidence that metalloprotease effector FocM35 seems to contribute to pathogen virulence by inhibiting the host immunity.

## 1. Introduction

Banana (*Musa* spp.) is one of the most important fruits and a staple diet for millions of people in the world [1]). The plant pathogen *Fusarium oxysporum* f. sp. *cubense* (*Foc*) causes Fusarium wilt on bananas, resulting in substantial yield losses. Four races of *Foc* have been classified based on different host banana types: race1, which is responsible for the Gros Michel epidemics, also attacks some AAB or ABB bananas; race 2 is pathogenic to cooking bananas such as ‘Bluggoe’ ABB; race 3 is no longer considered as *Foc* race since it only affects *Heliconia* species [2]; *Foc* race 4 is pathogenic to race 1- and race 2-susceptible cultivars as well as the Cavendish cultivars (AAA), which are grown and consumed worldwide nowadays. *Foc* race 4 isolates are further divided into two groups: subtropical race 4 (STR4) and tropical race 4 (TR4) [1]. *Foc* TR4 is more virulent and has spread from the Asia-Pacific region to the Middle East even to Latin America, as reported recently [1,3,4,5]. No efficient control measures have been developed against *Foc* TR4, other than the replacement of susceptible with resistant bananas or genetically modified Cavendish plants [6].

*Foc* TR4 is considered a hemibiotrophic pathogen [7]. Hemi-biotrophic pathogens deliver apoplastic and intracellular effector proteins to manipulate the host physiological metabolism and to induce cell death in order to facilitate pathogen colonization [8,9]. Integrating genome sequencing and bioinformatics, a large number of effector candidate genes have been predicted in *Foc* TR4 (data not published). However, functional analysis of predicted effector candidate genes has not been performed. So far, only a few effectors in *Foc* TR4 have been reported to be essential for virulence. For instance, FocSix1 and FocSix8, which are homologs of SIX (secreted in the xylem) effectors identified in *F. oxysporum* f. sp. *lycopersici* (*Fol*), are required for full virulence [10,11]. FocCP1 was demonstrated as a virulence factor that is required by *Foc* TR4 for penetration and full virulence [12]. However, none of these studies demonstrated the underlying molecular mechanisms involved in the host-pathogen interactions.

Metalloproteases of diverse families have been identified in various microbial pathogens with different functions in virulence contribution [13]. Among them, M35 (deuterolysin) and M36 (fungalysin) families are the two major types of metalloproteases secreted by pathogenic fungi. *Fusarium verticillioides* secretes Fv-cmp, a fungalysin protein with an M36 domain, to target and cleave class IV chitinases of maize at the chitin-binding domain [14]). An M36 metalloprotease UmFly of *Ustilago maydis* functions as a virulence factor by cleaving maize chitinase ZmChiA to promote fungal infection [15]. Several other plant pathogenic fungal, including *Colletotrichum higginsianum*, *Verticillium* species, and *Botrytis cinerea*, were also reported to cleave plant class IV Chitinase [16,17,18]. However, the functional roles of M35 family metalloprotease in the plant pathogenic fungi and their targets in plant remain largely unknown. M35 and M36 metalloproteases share the HEXXH motif [19], indicating M35 metalloprotease maybe also able to target class IV chitinases.

In this study, based on an earlier transcriptomics study on the infection process of banana roots by *Foc* TR4 (data not published), a potential secreted metalloprotease FocM35_1 was identified. It belongs to the M35 metalloprotease family that is conserved among various pathogens. It was demonstrated that FocM35_1 is a virulence factor of *Foc* TR4 and is able to decrease banana chitinase activity. Also, it was found that FocM35_1 was able to trigger cell death and suppressed the INF1-induced hypersensitive response in the nonhost plant *Nicotiana. benthamiana*. The cell death triggering effect of FocM35_1 was dependent on the predicted enzymatic site and was able to enhance the disease development during *Foc* TR4 infection.

## 2. Results

### 2.1. FocM35_1 Encodes a Metalloprotease That Is Conserved among Different Fusarium Strains

FocM35_1 was first identified from the RNA-Seq profiling of banana roots infected with *Foc* TR4 (data not published). Among the candidate effectors identified in II5 (specified *Foc* TR4 strain) genome, FocM35_1 showed continuous high expression in the transcriptome analysis. It encoded a putative protein with 351 amino acid residues and was predicted to contain an N-terminal signal peptide (SP) and an M35 domain, which has two zinc-binding histidines and catalytic residue glutamate in a HEXXH motif (Figure 1a).

A BlastP search against the National Center for Biotechnology Information (NCBI) non-redundant (nr) database identified 21 additional FocM35_1 homologs in a range of additional *Fusarium* species. Multiple sequence alignment of these M35 domain-containing proteins showed that they all contain a HEXXH motif, which is typical of the M35 protein family (Appendix A). Except for *Fusarium graminearum* and Fo5176, two orthologues of M35 protein were identified in *Fusarium* species that clustered into two paralogous clades (Figure 1b). Although both clade I and clade II *Fusarium* M35 proteins were orthologues of P46076 (M35 protein of *Aspergillus oryzae*) [20], members of clade I shared more sequence similarities with P46076. M35-domain containing proteins in clade II are clustered closer with *Magnaporthe* Avr-Pita effector (Figure 1b), suggesting they may have similar functions.

To confirm the secretory function of the predicted SP of FocM35_1, we tested the ability of the SP to direct the secretion of invertase using a yeast secretion system. Similar to Avr1b with a known functional signal peptide [21], the predicted signal peptide of FocM35_1 was able to mediate the secretion of invertase after being fused with the invertase (Appendix A), indicating FocM35_1 is indeed a secreted protein.

### 2.2. FocM35_1 Contributes to Foc TR4 Virulence

Expression levels of FocM35_1 were analyzed with qRT-PCR using samples from conidia, vegetative mycelium, and infected roots at different time points of infection. It was found FocM35_1 was slightly induced at 12 h post-inoculation (hpi) and highly induced at 48 hpi (Figure 2). Therefore, we postulated that FocM35_1 may play a role in the virulence of *Foc* TR4 during early infection stages on a banana.

To determine the virulence contribution of FocM35_1 in *Foc* TR4, a FocM35_1-deletion mutant (∆FocM35_1) was obtained by replacing the FocM35_1 region with hygromycin-resistance gene (HPH) in *Foc* TR4 reference strain II5 (Appendix A). Gene replacement mutants were preliminarily identified via PCR screening (Appendix A) using primer pairs listed in Appendix A. The candidate transformant was further confirmed by Southern blot analysis using a HPH-specific probe (Appendix A). Genetic complementation of the ∆FocM35_1 mutant (∆FocM35_1-C) was obtained by introducing into the ∆FocM35_1 strain a construct containing the FocM35_1 gene under the FocM35_1 native promoter. The complemented strains were also validated by PCR amplification (Appendix A).

The knockout mutant ∆FocM35_1 showed no differences in vegetative growth on potato dextrose agar (PDA) compared to the wild-type (WT) strain and the complementation strain ∆FocM35_1-C (Figure 3a,b). We further tested the virulence of ∆FocM35_1, ∆FocM35_1-C, and WT by root infection assays. Compared with WT and ∆FocM35_1-C, ∆FocM35_1 mutant caused significantly fewer symptoms on banana corms 30 days after inoculation of the conidia suspensions (Figure 3c). Of the banana plantlets inoculated with WT and ∆FocM35_1 mutant, 83.3% and 6.7% resulted in slight browning symptoms and no symptoms in corms, respectively. To determine whether the targeted disruption of FocM35_1 affected the fungal growth in planta, the ratio of fungal DNA to banana DNA was measured using quantitative PCR (qPCR). The biomass of ∆FocM35_1 mutant in banana corms was significantly lower than that of WT (Figure 3e). These results suggest FocM35_1 plays an essential role during the banana root infection by *Foc* TR4.

In order to explore the underlying mechanisms of FocM35_1 in virulence contribution, the ∆FocM35 mutant strain for its ability to penetrate into cellophane membranes and banana roots was measured. The growth of fungi on plates covered with cellophane could be used as a condition to mimic the penetration phase in vitro [22]. It was found that the ability of the ∆FocM35 mutant to penetrate cellophane was significantly reduced compared to the WT strain (Appendix A). Consistently, the events of adhesion and penetration on a banana root surface by ∆FocM35 were also significantly reduced compared to the WT (Appendix A). These observations suggest that the lack of FocM35_1 probably led to a decreased fungal growth within host tissues.

It has been reported that tolerance against oxidative stress and osmotic stress is associated with the virulence of pathogenic fungi [23,24]. We, therefore, measured the sensitivity of ∆FocM35_1 and WT strains to oxidative stress on minimal medium (MM) plates supplemented with 20 mM H_2_O_2_ and to osmotic stress on MM plates supplemented with 1.5 M NaCl. The deletion mutant ∆FocM35_1 exhibited a significant decrease of tolerances to both oxidative stress mediated by H_2_O_2_ and osmotic stress mediated by NaCl, compared to that of the WT (Appendix A). Taken together, these results indicated that the ∆FocM35 mutant strain was sensitive to both oxidative and osmotic stress, which may have contributed to its loss of virulence.

### 2.3. FocM35_1 Could Decrease Banana Chitinase Activity and Target at MaChiA

It was reported that M36 metalloprotease was able to target and cleave class IV plant chitinase [15,16]. We investigated whether FocM35_1 could influence banana chitinase activity. FocM35_1 protein was expressed in *E. coli* and purified (Appendix A). Then the recombinant His-FocM35_1 was applied to banana embryogenic cell suspensions (ECSs). Treatment with His-FocM35_1 protein significantly decreased banana chitinase activity dose-dependently (Figure 4a). As a control, chitinase activity of ECSs incubated with BSA or elution buffer remained unchanged (Figure 4a).

To test whether FocM35_1 directly interacts with chitinase, we first analyzed the expression patterns of genes encoding banana chitinases with GH19 (glycosyl hydrolase family 19) domain during the infection of Cavendish banana by *Foc* race 1 and *Foc* TR4, respectively (data not published). Based on the expression patterns obtained from the RNA-Seq data, these genes were classified into two large clusters (Appendix A). Four genes in cluster II (Ma05_17850, Ma03_28030, Ma06_31980, and Ma09_20710) are highly expressed during the pathogen infection (Appendix A). Since Ma09_20710 showed higher homology with ZmChiA [15] than the other members in this chitinase family, it was designated as MaChiA and used to investigate its interaction with FocM35_1. Biomolecular fluorescence complementation (BiFC) assay was performed in rice protoplasts to test the interaction between FocM35_1 and MaChiA. FocM35_1-nsp (with no signal peptide) and MaChiA fragments were fused with the split N-terminal and C-terminal in the vectors of pRTVnVN and pRTVnVC, respectively. After co-transfection, the ECFP and mCherry fluorescence indicated successful transfection and expression of the two plasmids. Then we detected clear yellow fluorescence signals on the plasma membrane of the transfected protoplasts (Figure 4b), indicating the formation of reconstituted mVenus through the FocM35_1-MaChiA interaction. Therefore, FocM35_1 seems able to bind the banana chitinase MaChiA directly.

### 2.4. The Metalloprotease Activity of FocM35_1 Is Required for Triggering Cell Death in N. benthamiana

To investigate the influence of FocM35_1 on plant immunity, we performed *Agrobacterium tumefaciens*-mediated transient expression of FocM35_1 in *N. benthamiana* leaves to test whether they trigger cell death. Empty vector and the oomycete PAMP (pathogen-associated molecular pattern) elicitor INF1 were also expressed as controls (Figure 5a). We found that FocM35_1 expression induced cell death in *N. benthamiana* leaves as well as the INF1 control, but not the empty vector control (Figure 5b). Next, we tested whether the signal peptide and the predicted catalytic site (amino acid position 303) of FocM35_1 are responsible for causing cell death in *N. benthamiana*. A signal peptide (1-15 bp)-deleted mutant (FocM35_1-nsp) and a catalytic site missense (303rd amino acid to alanine) mutant were constructed (Figure 5a) and transiently expressed in *N. benthamiana*. We found that FocM35_1-nsp and FocM35_1^E303A^ failed to induce cell death (Figure 5b). Immunoblotting analysis confirmed that all the corresponding proteins were successfully expressed in *N. benthamiana* leaves (Figure 5c), except for FocM35_1-nsp. Cell death induction activities were determined by ion leakage measurement and were consistent with the cell death phenotypes (Figure 5d). Furthermore, it was found that FocM35_1 aggregated at the apoplastic space of *N. benthamiana* cells through subcellular localization studies (Figure 5e). These results suggested that FocM35_1 is secreted into the apoplast and triggers cell death in the host plant. In addition, the signal peptide, and the enzymatic activity of FocM35_1 are required for its cell death-promoting function.

### 2.5. FocM35_1 Suppresses INF1-Induced Cell Death in N. benthamiana and Accelerates Infection Process of Foc TR4

We next investigated whether FocM35_1 can suppress plant immune responses. Since the oomycete PAMP elicitor INF1 strongly induces cell death in *N. benthamiana* [25], we examined the possible effect of FocM35_1 on the INF1-elicited immune response in *N. benthamiana*. Transient expression of INF1 by agroinfiltration induced rapid HR cell death in *N. benthamiana* within 3 d post-inoculation (dpi), which is faster than FocM35_1-induced cell death that appeared at 5 dpi (Figure 6a). Furthermore, it was found that when *N. benthamiana* leaves were pre-infiltrated with Agrobacterium harboring a plasmid expressing FocM35_1, the INF1-induced cell death in the same leaf was significantly delayed from 3 dpi to 6 dpi (Figure 6a). Moreover, the mutant FocM35_1^E303A^ failed to suppress the INF1-induced cell death, indicating the enzymatic activity of FocM35_1 is required for the suppression of INF1-induced cell death (Figure 6a). Western blot analysis showed that INF1 was co-expressed with GFP, FocM35_1, and FocM35_1^E303A^ in *N. benthamiana* leaves (Figure 6b). Electrolyte leakage in *N. benthamiana* triggered by INF1 was significantly attenuated in the presence of FocM35_1 (Figure 6c). Collectively, these data suggest that FocM35_1 suppresses INF1-induced cell death in a plant.

To further examine the biological function of FocM35_1, banana leaves were inoculated with the recombinant FocM35_1 protein before the inoculation of *Foc* TR4. It was found the lesion formation in banana leaves was greatly enhanced by this co-inoculation of recombinant FocM35_1 and *Foc* TR4 (Figure 7). These findings suggest that FocM35_1 enhances the *Foc* TR4 infection by suppressing the plant immune response to promote the lesion formation, and the suppression is likely targeting the INF1-induced HR response.

## 3. Discussion

To date, few virulence effector proteins have been identified to be essential for *Foc* TR4 virulence. Taking advantage of the well-annotated genomic sequence of *Foc* TR4, hundreds of effector-coding genes were predicted (data not published). Further analysis of the transcriptome of *Foc* TR4 infection processes revealed many effector candidates whose expressions are highly induced during different infection stages (data not published). One of the putative effector proteins, FocM35_1, was predicted to function as a metalloprotease as it contains the conserved HEXXH motif. Expression pattern analysis based on qRT-PCR and previous transcriptome profiling showed that FocM35_1 was highly expressed during the early stage of infection. The knockout of FocM35_1 significantly reduced *Foc* TR4 virulence in banana plantlets. In this study, we verified that FocM35_1 contains a predicted signal peptide, and its secretion was verified using a yeast secretion system. In addition, subcellular localization assays showed that FocM35_1 was mainly accumulated in the apoplast, indicating that FocM35_1 is secreted into the apoplast region by *Foc* TR4 and plays a role in pathogenesis.

Metalloproteases have been implicated as important virulence factors in both bacterial and fungal pathogens [15,26,27]. Previous studies described that an M36 metalloprotease FvFly1 of *F. verticillioides* cleaves ZmChiA at the hinge domain to release the chitin-binding domain and the hydrolase domain as byproducts [14]. Another M36 metalloprotease, FoMep1 of *Fol*, was also demonstrated to target the same hinge domain site of tomato SlChi1 [17]. In addition, FoMep1 showed a synergized function with the serine protease FoSep1 to influence the pathogenesis of *Fol* [17]. Although how M36 metalloproteases modulate the host IV class chitinases activity during pathogenic development was well-studied, it is still unknown how M35 metalloproteases of phytopathogens interact with host targets. M35 metalloproteases share the same HEXXH enzyme domain with M36 metalloproteases [19] and may also target chitinase under a similar mechanism. Therefore, this study tested and showed that FocM35_1 was able to target banana chitinase MaChiA and decrease chitinase activity, and deletion of FocM35_1 reduced virulence of *Foc* TR4, indicating the importance of FocM35_1 in pathogenesis through its interaction with plant chitinase. Whether FocM35_1 could cleave MaChiA remains to be further studied in the future. We also found another M35 metalloprotease (FocM35_2) in the *Foc* genome, but whether FocM35_2 is also involved in fungal virulence should be determined in the future.

At the same time, we performed a range of agroinfiltration assays and found FocM35_1 was able to trigger cell death in *N. benthamiana* leaves. We also found the ability of FocM35_1 to trigger cell death depends on its signal peptide and metalloprotease activity. We propose that FocM35_1 is secreted into extracellular spaces, targeting components of plant immunity that further trigger downstream cell death. Cell death is considered a defense response, as it can prevent pathogens’ further invasion by eliciting host immune response [28]. However, in hemi-trophic and necrotrophic pathogens, cell death induced by some effectors and toxins is also regarded as a strategy to facilitate the invasion of pathogens [29]. *Phytophthora parasitica* is a hemi-trophic phytopathogen, and its cysteine proteases PpCys44 and PpCys45 positively promote pathogen virulence during infection by triggering plant cell death [30]. Here, we reported FocM35_1 induced cell death in *N. benthamiana* leaves and suppressed the INF1-induced cell death. INF1 is a well-known oomycete PAMP elicitor, which strongly induces cell death in *N. benthamiana* and triggers plant immune defenses [25]. One possible explanation for how FocM35_1 suppressed INF1-induced cell death while still induced cell death by itself in *N. benthamiana* is that FocM35_1 may target and cleave a receptor of INF1, blocking the INF1-induced cell death, while FocM35_1-triggered cell death may be deployed by a pathogen to enhance virulence and promote fungal invasion. A similar mechanism was reported on the effector CoNIS1 of *Colletotrichum orbiculare* [31]. CoNIS1 interact with PRR-associated kinases BAK1 and BIK1, which are the receptors of INF1, thereby inhibiting INF1-induced cell death [31]. To better elucidate the underlying mechanism of FocM35_1 in suppression of the INF1-associated hypersensitive response, future studies should be performed to investigate the interaction between FocM35_1 and INF1-related plant defense pathway.

Interestingly, pretreatments of recombinant FocM35_1 promoted the invasion of *Foc* TR4, indicating FocM35_1 induces cell death to further enhance fungal virulence. Since *Foc* TR4 is a typical hemi-biotrophic plant pathogen, cell death may facilitate pathogen infection. It implied that FocM35_1 may play an important role at an early stage of infection by promoting the transition from biotrophy to necrotrophy lifestyle. Overall, the results of this study not only revealed the pathogenesis mechanism of a new virulence effector of *Foc* TR4 but also demonstrated the significant role of M35 family metalloprotease in fungal virulence.

## 4. Materials and Methods

### 4.1. Plant Growth Conditions, Fungal Strains and Growth Conditions

‘Cavendish’ banana (AAA) cv ‘Brazilian’ plantlets with 5 to 6 leaves (*c*. 20 cm height) were grown in a temperature-controlled glasshouse (14 h light and 10 h dark cycle, at 28 °C with 40% humidity). *N. benthamiana* was grown in the growth chamber at 25 °C. The wild-type (WT) and mutants of *Fusarium oxysporum* f. sp. *cubense* TR4 strain II5 were cultured on potato dextrose agar (PDA) or in potato dextrose broth (PDB) at 28 °C.

### 4.2. Bioinformatics Analysis

Gene ID of FocM35_1 in NCBI is 42027094. A search of M35 metalloprotease proteins in 12 *Fusarium* species was performed using a BlastP search on the NCBI non-redundant (nr) database. ClustalX 2.1 [32] was used for protein alignment and, ESPript 3.0 [33] was used for drawing alignment graphs with secondary structure information. A phylogenetic tree was constructed using MEGA X [34] maximum likelihood method with 1000 bootstrap replications. SignalP v. 4.0 [35] was used for signal peptide prediction. The expression patterns of banana chitinases were based on the RNA-seq data for Cavendish banana infected with *Foc* race 1 and *Foc* TR4, respectively, at 18, 32, 56 h post-inoculation (hpi) (data not published). The heatmap of all chitinase-coding genes was generated with Tbtools [36] using fragments per kilobase of transcript per million mapped reads (FPKM) values of each gene.

### 4.3. Signal Peptide Secretion Test

The function of predicted signal peptides of FocM35_1 was evaluated according to a previously published protocol [37]. Briefly, the predicted signal peptide of FocM35_1 was fused into pSUC2 vector and, then the constructs were transformed into yeast *Saccharomyces cerevisiae* YTK12 strain. All transformants were cultured on CMD–W medium (6.7 g/L yeast nitrogen base without amino acids, 0.74 g/L -Trp DO supplement, 20 g/L sucrose, 1 g/L glucose, and 20 g/L Agar A) to select positive colonies. Colorimetric change (to red) after adding 0.1% TCC solution is observed if the signal peptide is functional.

### 4.4. Construction of Gene Replacement and Complementary Strain

The ΔFocM35_1 mutant strain was constructed by homologous recombination as previously described [12]. Briefly, the 5′- and 3′-flanking sequences of the FocM35_1 gene were amplified using the genomic DNA of II5 as a template with the Phanta Max polymerase (Vazyme, China). The primers used in this study were listed in Appendix A. The FocM35_1 gene was replaced by a hygromycin-resistance cassette (HPH), which was driven by a constitutive TrpC promoter amplified from the pCT74 vector. PCR products were transformed into protoplasts of WT strain II5 by protoplast transformation mediated by polyethylene glycol (PEG). The putative deletion mutants were identified by PCR analysis and further confirmed by Southern blot. Gene complementation vector was constructed by introducing native promoter-FocM35_1 fragment into pYF11 (Geneticin resistance) vector, and the complemented strain was also verified by PCR amplification.

### 4.5. Phenotyping, Stress Sensitivity and Cellophane Membrane Assays

In order to determine whether there is any phenotypic difference between WT, ΔFocM35_1, and ΔFocM35-C, 5 mm mycelial plug of each strain was inoculated on PDA plate for five days in the incubator at 28 °C. To determine the difference in stress responses between WT and mutants, strains were cultured on PDA plates supplemented with the final concentrations of 1.5M NaCl or 20 mM H_2_O_2_ for five days at 28 °C. In addition, the penetration ability of each strain was tested on the cellophane membrane as described previously [12]. The experiments were repeated three times.

### 4.6. Pathogenicity Assays and Fungal Biomass Estimation

‘Cavendish’ banana (AAA) cv ‘Brazilian’ plantlets with 5 to 6 leaves (*c*. 20 cm height) were inoculated with *Foc* strains conidial suspensions at a concentration of 1 × 10^7^ conidia L^−1^, with clear water as the negative control. In order to evaluate the pathogenicity, the disease severity of each experimental plant was evaluated as previously reported [38].

Quantitative PCR (qPCR) was performed to determine the fungal biomass in the rhizome [38]. Elongation factor 1-α (FocEF1α) gene of *Foc* TR4 and banana actin (MusaActin) gene were used to quantify fungi biomass and plant, respectively. (Appendix A)

### 4.7. RNA Extraction and Quantitative RT-PCR Analysis

Total RNA of infected banana roots was extracted using a SteadyPure Plant RNA Extraction Kit (Accurate Biology, Hunan, China) following the manufacturer’s instructions. Genomic DNA was eliminated, and reverse transcription reaction was carried out using an Evo M-MLV RT Kit (Accurate Biology, Hunan, China). To analyze the expression pattern of FocM35_1 during infection, qRT-PCR analysis was performed in Step Two real-time PCR system (Accurate Biology, Hunan, China) with the SYBR^®^ Green Premix (Applied Accurate Biology). FocEF1α was used as an internal control to normalize the data and, the relative expressions of genes were calculated using the 2^−ΔΔCT^ method. The gene-specific primers used for qRT-PCR are listed in Appendix A.

### 4.8. Scanning Electron Microscopy Observation

For SEM observation, ‘Cavendish’ banana (AAA) cv ‘Brazilian’ plantlets with 5 to 6 leaves (*c*. 20 cm height) were inoculated with *Foc* strains conidial suspensions at a concentration of 1 × 10^7^ conidia/L. The root samples were collected 48 h later and observed using a Hitachi Model S-3400N scanning electron microscope (Hitachi, Tokyo, Japan).

### 4.9. Protein Expression

The cDNA of FocM35_1 was amplified and cloned into the expression vector pET-28a-His for expression in *E. coli* Rosetta (DE3). Transformed cells were cultured overnight at 37 °C in LB medium and induced with isopropyl β-D-1-thiogalactopyranoside (IPTG) for 24 h at 28 °C. The cells were lysed by supersonic treatment, and the supernatant from the lysate was applied to a Ni-NTA column (Transgene Biotech, Beijing, China). Protein concentration was determined using the microtiter plate method (Sangon Biotech, Shanghai, China) and was also confirmed by SDS-PAGE gel stained with Coomassie Blue.

### 4.10. Chitinase Activity Assays

The effect of FocM35_1 on banana chitinase was determined on banana ECSs. ECSs were incubated with purified His-FocM35_1 protein of different concentrations (5, 10, 20 μM) at 28 °C for 1 h. The control treatment included equal concentrations of BSA or elusion buffer alone. The protein isolation of banana ECSs was performed as previously described [38].

Chitinase activity was determined using 40 mM 4-Nitrophenyl β-D-*N*,*N*′,*N*′′-triacetylchitotriose (Sigma, St. Louis, MO, USA) as a substrate. 10 mL ECSs extract at 0.2 mg/mL was mixed with an equal volume of substrate solution and incubated at 37 °C for 1 h. Then the reaction was quenched by the addition of 20 mL of 1 M Gly-NaOH (pH 10.2). The release of the chromophore ρ-nitrophenol (pNP) was measured at 405 nm. One unit (U) of activity per mg chitinase (U/mg) was defined as the release of 1 mmol of pNP/mg per minute under the assay conditions.

### 4.11. Rice Protoplasts Isolation and Transfection for BiFC (Bimolecular Fluorescence Complementation) Assay

The interaction between FocM35_1 with MaChiA was verified using the BiFC system. FocM35_1 and MaChiA gene was introduced into pRTVnVN and pRTVnVC vectors, respectively. The isolation, transfection, and fluorescence detection were carried out according to the method described before [39].

### 4.12. Subcellular Localization Assay

To determine the subcellular localization of FocM35_1, *A. tumefaciens* EHA105 containing the vector pCAMBIA1300 with FocM35_1 was transiently expressed in *N. benthamiana* in an infiltration medium with an OD_600_ of 1.0. EHA105 carrying the empty pCAMBIA1300 vector was used as control.

At two days post-infiltration, epidermal tissues of tobacco leaves were sampled and observed with a laser confocal microscope (LSM 710, Carl Zeiss, Oberkochen, Germany).

### 4.13. Agroinfiltration Assays

The constructed vectors pCAMBIA1300 carrying FocM35_1 (with or without signal peptide) or, FocM35_1^E303A^ (catalytic site mutation) were transformed into *A. tumefaciens* EHA105, respectively. Overnight-cultured recombinant strains of *A. tumefaciens* cells were infiltrated into the leaves of 6-week-old *N. benthamiana* plants, and the same Agrobacterium strain harboring the empty pCAMBIA1300 vector was used as a negative control, with INF1 as a positive control.

In order to test whether FocM35_1 could suppress INF1-induced cell death, at one day after infiltration with *A. tumefaciens* carrying FocM35_1, the infiltration site was further infiltrated with a recombinant *A. tumefaciens* carrying FocM35_1 or FocM35_1^E303A^. All tests were performed three times.

The total proteins of *N. benthamiana* leaves were extracted with a plant protein extraction kit (Sangon Biotech, Shanghai, China) according to the manufacturer’s instruction. Then the proteins were analyzed by SDS-PAGE and immunoblotting.

### 4.14. Banana Leaf Infection

Leaves of ‘Cavendish’ banana (AAA) cv ‘Brazilian’ plantlets were treated with recombinant FocM35_1 or elution buffer (phosphate buffer saline, pH 7.0). One hour after treatment, *Foc* spore suspension was infiltrated into the same area of leaves. Disease phenotype was observed at 5 d post-inoculation. The experiments were repeated three times.

## Figures and Tables

**Figure 1 pathogens-10-00670-f001:**
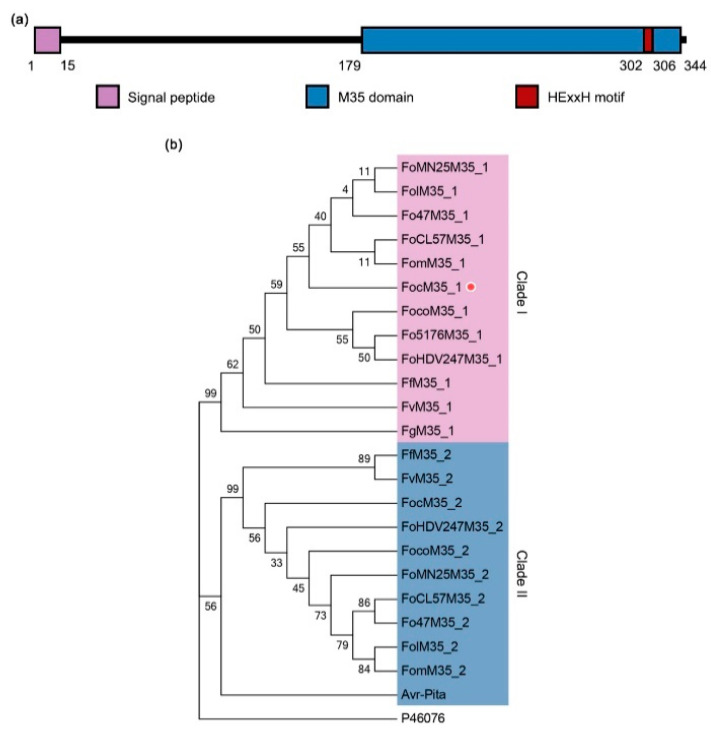
Sequence analysis of FocM35_1. (**a**) Schematic structure of the FocM35_1 protein, a M35 metalloprotease. (**b**) Maximum likelihood tree based on P46076 (M35 protein of *Aspergillus oryzae*) and Avr-Pita and their orthologous proteins from *Fusarium* spp. genomes. Red dot indicates FocM35_1.

**Figure 2 pathogens-10-00670-f002:**
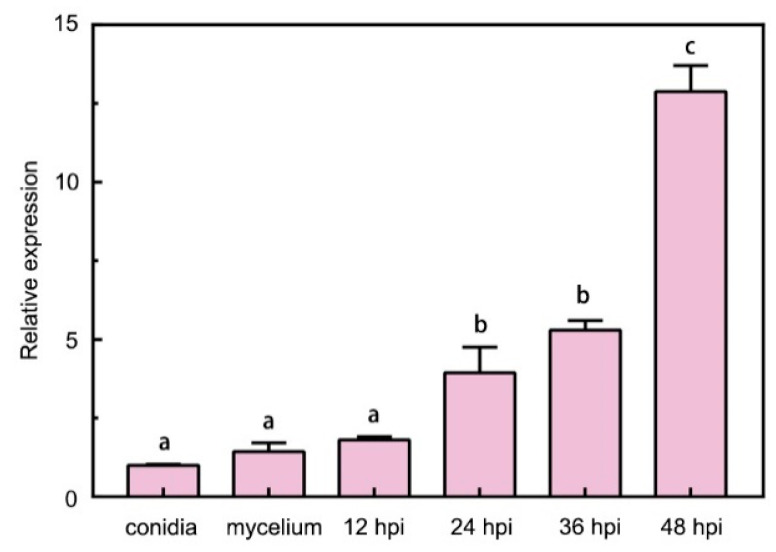
FocM35_1 expression at different development stages. The relative expression levels were compared to that of the conidia (set as 1). The fungal constitutive gene FocEF1α was used as internal reference. Data are the means of tree independent experiments. The letters above the bars indicate statistically significant differences at *p* < 0.05 (Student’s *t*-test).

**Figure 3 pathogens-10-00670-f003:**
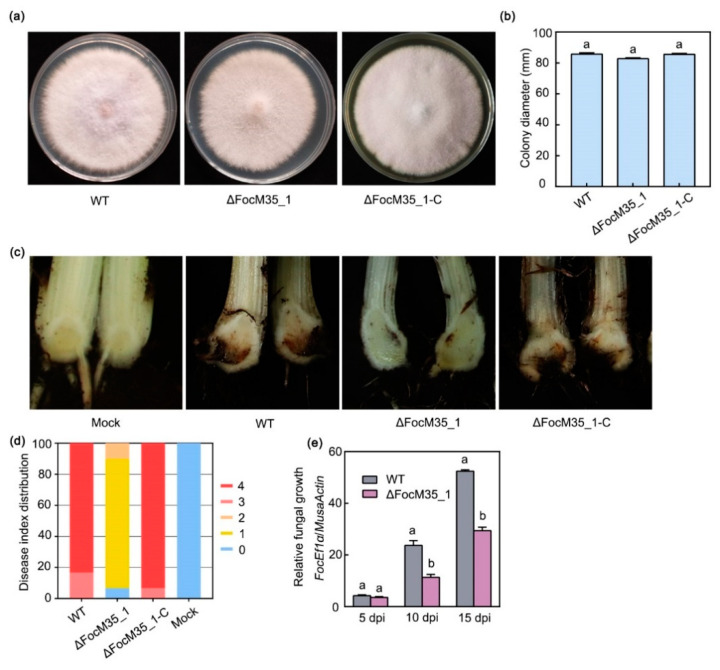
FocM35_1 contributes to the virulence of *Foc* TR4. (**a**) Colony morphologies of wild type (WT), ∆FocM35_1 and ∆FocM35_1-C strains cultured on potato dextrose agar (PDA) plates. Photos were taken 5 days after incubation at 28 °C. (**b**) Colony diameter of the WT, ∆FocM35_1 and ∆FocM35_1-C strains shown in a. (**c**) Disease phenotype and (**d**) disease index distribution in banana plantlets inoculated with the WT, ∆FocM35_1 and ∆FocM35_1-C strains at 30 days post inoculation (dpi). (**e**) Fungal growth in banana roots, determined by quantitative polymerase chain reaction (qPCR). Data presented in (**b**) and (**e**) are means ± SDs from three independent experiments. The letters above the bars indicate statistically significant differences at *p* < 0.05 (Student’s *t*-test).

**Figure 4 pathogens-10-00670-f004:**
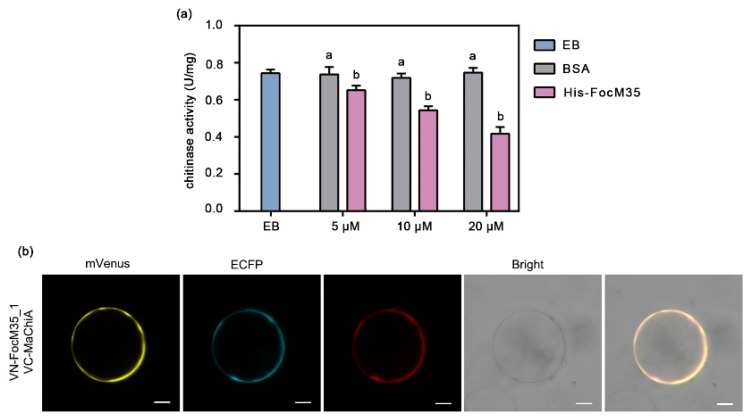
Metalloprotease activity of FocM35_1. (**a**) Treatment with His-FocM35_1 protein decreased chitinase activity. Banana embryonic suspension cells were incubated with elusion buffer (EB), different concentrations of BSA or His-FocM35_1 protein individually. Chitinase activity was assayed. All data are means ± SDs from three independent experiments. Different letters above the histograms indicate statistically significant differences at *p* < 0.05 (Student’s *t*-test). (**b**) FocM35_1 and MaChiA interaction detected in rice membrane by bimolecular fluorescence complementation (BiFC) assay. The mVenus channel exhibit Venus fluorescence reflecting the direct interaction of FocM35_1 and MaChiA. Bars = 10 μm.

**Figure 5 pathogens-10-00670-f005:**
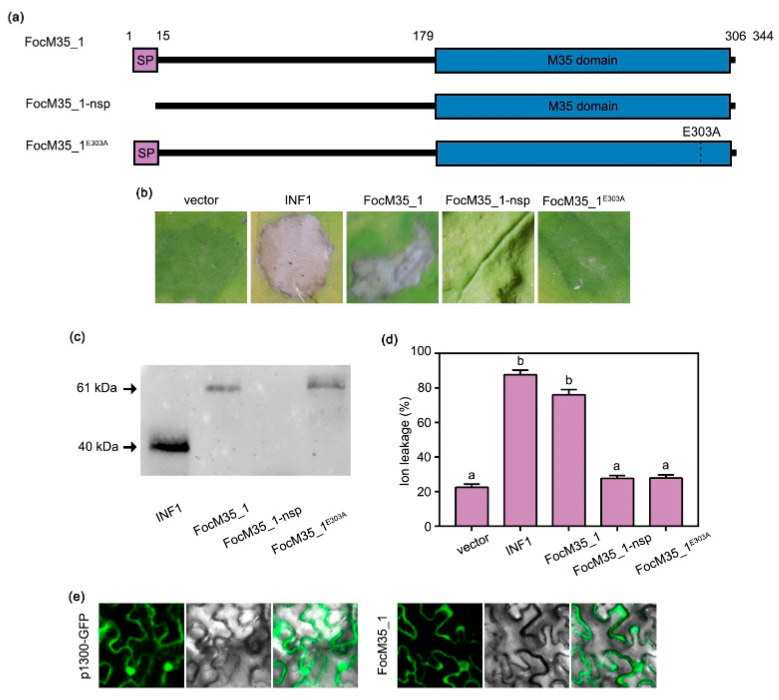
The signal peptide and enzymatic activity of FocM35_1 is required for its cell death-promoting effects. (**a**) Schematic illustration of FocM35_1 deletion and point mutation mutants used in this study. (**b**) *Nicotiana benthamiana* leaves were infiltrated with *Agrobacterium tumefaciens* carrying vector, INF1, FocM35_1 and its mutants. Photographs were taken at 5 dpi. Similar results were obtained in two additional experiments. (**c**) Western blotting detection of control, INF1, FocM35_1 and its mutants using anti-GFP. (**d**) Quantification of cell death by electrolyte leakage measurement. Different letters above the histograms indicate statistically significant differences at *p* < 0.05 (Student’s *t*-test). (**e**) Subcellular localization of FocM35_1 by transient expression of green fluorescent protein (GFP)-fused FocM35_1 in *N. benthamiana* leaves. Photographs were taken at 2 dpi. The vector was used as control.

**Figure 6 pathogens-10-00670-f006:**
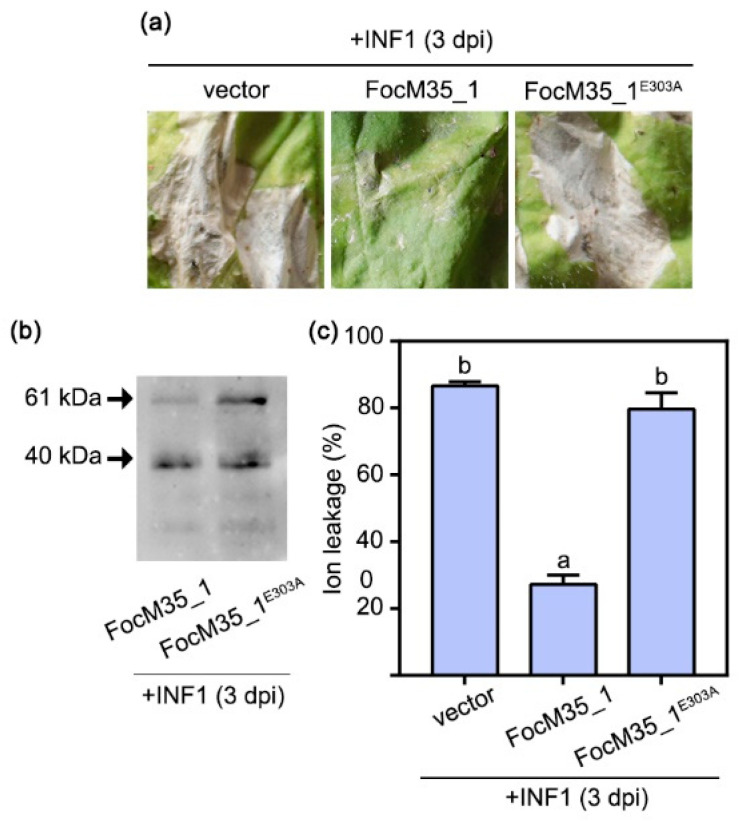
FocM35_1 suppress INF1-induced cell death and its enzymatic activity is required for immunosuppressive ability. (**a**) 1 day after infiltration with *A. tumefaciens* carrying vector, FocM35_1 and FocM35_1-nsp (without signal peptide), *N. benthamiana* leaves were further challenged with *A. tumefaciens* carrying INF1 at the same sites. The photographs were taken at 3 dpi. Similar results were obtained in two additional experiments. (**b**) Western blotting detection of FocM35_1 and INF1 using anti-GFP. (**c**) Quantification of cell death by electrolyte leakage measurement. Different letters above the histograms indicate statistically significant differences at *p* < 0.05 (Student’s *t*-test).

**Figure 7 pathogens-10-00670-f007:**
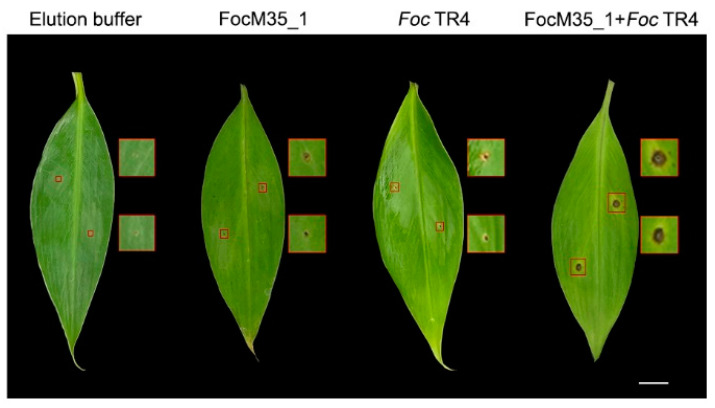
FocM35_1 promoted lesion formation of *Foc* TR4 in banana leaves. Banana leaves were infiltrated with elution buffer, recombinant FocM35_1, *Foc* TR4 or recombinant FocM35_1 together with *Foc* TR4. Photographs were taken 4 dpi. The areas within the red squares are enlarged into pictures at the right side. Bar = 1 cm.

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
