# Peer review of "The M35 Metalloprotease Effector FocM35_1 Is Required for Full Virulence of Fusarium oxysporum f. sp. cubense Tropical Race 4"

_pathogens, 2021, doi:10.3390/pathogens10060670_

Round 1

Reviewer 1 Report

For the comments, see the attached file, please.

Author Response

Thank you for your constructive comments concerning our article. These comments are all valuable and helpful for improving our article. We have submitted a revised manuscript with changes highlighted, and here we provided a point-to-point response to review below for your consideration.

Line 76 : « triggering » rather than « formation »

Answer: Thank you for pointing this out and we have changed the “formation” to “triggering”. (Line 82)

Line 82 : specify strain II5

Answer: Thanks. We have corrected it accordingly. (Line 87-88)

Lines 119-125 : A detailed legend of the figure S3 has to be provided with references to the a, b, c and d parts.

Answer: Thank you for pointing this out. We have added the captions of supplementary materials in the newly uploaded supplementary files.

Line 149 : only 1 WT strain ?

Answer: Thank you for pointing this out. We have obtained several mutants, and the mutants had the same phenotype under these cultural conditions, thus we only showed one representative result.

Line 149-160 : The relevant parts of the figure S4 have to be precised.

Answer: Thank you for pointing this out. We have added the captions of supplementary materials in the newly uploaded supplementary files.

Line 170 : « elution » instead of « elusion » , I guess.

Answer: Thanks. We have corrected it accordingly. (Line 180)

Line 172 : « GH19 domain » has to be specified.

Answer: Thanks. We have specified the GH19 domain. (Line 183-184)

Line 191 : The legend of the figure 4b has to be completed (for exemple, ECFP ?...).

Answer: Thanks. We have added the relative information in the text. (Line 194-196)

Line 198 : PAMP has to be defined.

Answer: Thanks. Full name of PAMP has been added. (Line 211-212)

Line 220 : It should be precised that the protein of interest is fused to the GFP coding sequence.

Answer: Thank you for pointing this out. We have corrected it accordingly. (Line 235-236)

Line 246 : The legend of the figure 6b) does not seem to be exact as the control and theFocM35_1-nsp lanes are not on the figure.

Answer: Thank you for pointing this out. We have corrected it accordingly. (Line 259-260)

Line 315 : in suppression of the INF1-associated instead of in suppression the INF1-associated.

Answer: Thank you for pointing this out. We have corrected it accordingly. (Line 331)

Line 335 : The query sequence should be given.

Answer: Thank you for pointing this out. We have added the gene ID. (Line 350)

Line 336 : ESPript instead of ESPint.

Answer: Thanks. We have corrected it accordingly. (Line 352)

Line 383 : « was carried out using » instead of « used ».

Answer: Thanks. We have corrected it accordingly. (Line 402-403)

Line 387 : control instead of controls.

Answer: Thanks. We have corrected it accordingly. (Line 406)

Line 430 : vectors instead of vector.

Answer: Thanks. We have corrected it accordingly. (Line 449)

Line 431 : were instead of was

Answer: Thanks. We have corrected it accordingly. (Line 450)

Reviewer 2 Report

The M35 metalloprotease effector FocM35_1 is required for full virulence of Fusarium oxysporum f. sp. cubense tropical race 4 “ “provides evidence that metalloprotease effector FocM35 28 seems to contribute to pathogen virulence by inhibiting the host immunity”.

The paper reports significant findings for a better understanding of the pathogenesis of banana Fusarium wilt, describing the work done in a precise and very technical way. Nonetheless, the excess of technicality (e.g., frequent use of acronyms even when it could be avoided) makes reading quite difficult. It is advisable to lighten the text by making it smoother.

As to the style, it is generally desirable to avoid the first person in sentences (I, we, my, our and so on) when writing scientific papers to maintain a neutral super part description of results. Furthermore, this style is adopted only in some sections and not in the whole paper. Please, uniformize the text.

Finally, the authors do not express any hypothesis on a possible use of the results for the fight against the disease: it would be useful to indicate whether these findings can have practical application in the field.

For the rest, there are some small inaccuracies which should be corrected:

line 43: Latin instead of Lattin

line 75: the extended name of Nicotiana benthamiana is expressed in the abstract, but it could be beneficial to define it also in the text at the first citation

line 157 and 159: please, amend H2O2 with H2H2

lines 121, 356, 379, 389: a “Table S1” is repeatedly cited in the text but is not attached. Please, provide it or delete the citations, or otherwise specify if it is comprised in the online “supplementary materials”

Figures: the figures are very small in dimension, so it is very difficult to correctly evaluate them at the size normally used to read text; it is advisable to provide larger formats. In addition, figure 3 is quite difficult to follow with the present sequence of 3a, 3b, 3c and so on. Please, rearrange them in a more ordinate organization.

Author Response

Thank you for your constructive comments concerning our article. These comments are all valuable and helpful for improving our article. Accordingly, we have modified several sentences to avoid the first-person description and added the full name of the acronyms. We have submitted a revised manuscript with changes highlighted, and here we provided a point-to-point response to review below for your consideration.

line 43: Latin instead of Lattin

Answer: Thanks. We have corrected it accordingly. (Line 43)

line 75: the extended name of Nicotiana benthamiana is expressed in the abstract, but it could be beneficial to define it also in the text at the first citation

Answer: Thank you for pointing this out. We have defined it again. (Line 81)

line 157 and 159: please, amend H2O2 with H2O2

Answer: Thanks. We have corrected it accordingly. (Line167 and 169)

lines 121, 356, 379, 389: a “Table S1” is repeatedly cited in the text but is not attached. Please, provide it or delete the citations, or otherwise specify if it is comprised in the online “supplementary materials”

Answer: Thanks. Table S1 is included in the “supplementary materials”.

Figures: the figures are very small in dimension, so it is very difficult to correctly evaluate them at the size normally used to read text; it is advisable to provide larger formats. In addition, figure 3 is quite difficult to follow with the present sequence of 3a, 3b, 3c and so on. Please, rearrange them in a more ordinate organization.

Answer: Thank you for pointing this out. We have corrected the Figure 3 accordingly and enlarged the figures in the newly uploaded paper.

Reviewer 3 Report

Review for Pathogens: 21 May 2021

Journal: Pathogens
Manuscript ID: pathogens-1213041
Type of manuscript: Article
Title: The M35 metalloprotease effector FocM35_1 is required for full virulence of Fusarium oxysporum f. sp. cubense tropical race 4
Authors: Xiaoxia Zhang, Huoqing Huang, Bangting Wu, Jianghui Xie, Altus Viljoen, Wei Wang, Diane Mostert, Yanling Xie, Gang Fu, Dandan Xiang, Shuxia Lyu, Siwen Liu, Chunyu Li

Summary: This study examines a metalloprotease effector gene from the important banana pathogen Fusarium oxysporum f. sp. cubense tropical race 4. The gene was identified from RNA-seq data (not presented here nor published) analysing gene expression of the pathogen during the early stages infection of banana roots. The authors performed a variety of analyses to identify and compare homologs including the signal peptide domain. The construction of a knock-out mutant and complementary strain has enabled comparison to the wild type and experimental confirmation of the necessity of this gene for Fusarium oxysporum f. sp. cubense tropical race 4 to infect banana. This is a very nice study, and it is presented in a very clear and well-written paper. I commend the authors for this. It was very interesting to read.

Novelty and impact of the study:

  • This study describes a novel effector from Fusarium oxysporum sp. cubense tropical race 4 and its requirement for infection of banana.
  • The authors show that the potential mode of action is by inhibition of host immunity.
  • Fusarium wilt of bananas is a significant disease on an important crop, and this study makes a significant contribution to the literature by identifying this virulence factor and the essential role it plays in this disease.

Possible improvements:

  • P4, line 131 – avoid starting sentence with number
  • P4, line 131 – This sentence needs some restructure and clarification...something like "Of the plants inoculated with wild type and knockout strains [state the names of strains] 83 and 6.7 % resulted in xxx & xxx, respectively". At the moment you can’t tell which strain caused which result/%.
  • Figure 3b legend: Colony diameters on PDA: some additional conditions should be stated e.g. after how long and at what temperature?
  • The layout of the sections of Figure 3 doesn't make sense. should be in order in way they are discussed and (a) to (e) left to right. They are mixed up.
  • P4, line 147 - I think a little more intro to the background to cellophane experiment would be helpful
  • P4, line 149 - what do you mean by both the wild type strains? DO you mean the wild-type and the complemented strain? In Figure S4 there is only one wild-type picture for each panel?
  • P4, line 149 - It would be good to indicate which panel in Figure S4 e.g. (Figure S4a) directly after presenting those results in the text. Be sure to present results for each in the text.
  • P5, line 160 - Figure S4 (d) - the y-axis label - should that just be Growth inhibition (%) not growth inhibition rate (%)?
  • P5, line 177 - Should this be Figure S6?
  • P7, line 224 - subtitle 2.5 - suppresses, accelerates
  • Figure citations are not required in the discussion. Check throughout.
  • Subtitle 4.1 - check comma

Author Response

 Thank you for your constructive comments concerning our article. These comments are all valuable and helpful for improving our article. We have submitted a revised manuscript with changes highlighted, and here we provided a point-to-point response to review below for your consideration.

P4, line 131 – avoid starting sentence with number

P4, line 131 – This sentence needs some restructure and clarification...something like "Of the plants inoculated with wild type and knockout strains [state the names of strains] 83 and 6.7 % resulted in xxx & xxx, respectively". At the moment you can’t tell which strain caused which result/%.

Answer: Thank you for pointing this out. We have changed the description accordingly. (Line 139-140)

Figure 3b legend: Colony diameters on PDA: some additional conditions should be stated e.g. after how long and at what temperature?

Answer: Thanks. We have added more information. (Line 149)

The layout of the sections of Figure 3 doesn't make sense. should be in order in way they are discussed and (a) to (e) left to right. They are mixed up.

Answer: Thank you for pointing this out. We have corrected the Figure 3 accordingly.

P4, line 147 - I think a little more intro to the background to cellophane experiment would be helpful

Answer: Thanks. We have added more information. (Line 157-158)

P4, line 149 - what do you mean by both the wild type strains? DO you mean the wild-type and the complemented strain? In Figure S4 there is only one wild-type picture for each panel?

Answer: Sorry for the mistake. It was only one WT. We have corrected it accordingly. (Line 160)

P4, line 149 - It would be good to indicate which panel in Figure S4 e.g. (Figure S4a) directly after presenting those results in the text. Be sure to present results for each in the text.

Answer: Thank you for pointing this out. We have corrected the description accordingly. (Line 160, 162, 170)

P5, line 160 - Figure S4 (d) - the y-axis label - should that just be Growth inhibition (%) not growth inhibition rate (%)?

Answer: Thank you for pointing this out. We have corrected the figure S4.

P5, line 177 - Should this be Figure S6?

Answer: Thank you for pointing this out. We have corrected it accordingly. (Line 188)

P7, line 224 - subtitle 2.5 - suppresses, accelerates

Answer: Thank you for pointing this out. We have corrected it accordingly. (Line 237)

Figure citations are not required in the discussion. Check throughout.

Answer: Thanks. We have deleted all the figure-citations in discussion.

Subtitle 4.1 - check comma

Answer: Thanks. We have corrected it accordingly. (Line 343)

This manuscript is a resubmission of an earlier submission. The following is a list of the peer review reports and author responses from that submission.